# Integrated Decision and Motion Planning for Highways with Multiple Objects Using a Naturalistic Driving Study

**DOI:** 10.3390/s25010026

**Published:** 2024-12-24

**Authors:** Feng Gao, Xu Zheng, Qiuxia Hu, Hongwei Liu

**Affiliations:** 1College of Mechanical and Vehicle Engineering, Chongqing University, Chongqing 400044, China; 202207021263t@stu.cqu.edu.cn (X.Z.); 20183201007@cqu.edu.cn (Q.H.); 2Western Science City Intelligent and Connected Vehicle Innovation Center (Chongqing) Co., Ltd., Chongqing 400015, China; liuhongwei@wicv.cn

**Keywords:** automated vehicle, motion planning, driving decision, artificial potential field, naturalistic driving study

## Abstract

With the rise in the intelligence levels of automated vehicles, increasing numbers of modules of automated driving systems are being combined to achieve better performance and adaptability by reducing information loss. In this study, an integrated decision and motion planning system is designed for multi-object highways. A two-layer structure is presented to decouple the influence of the traffic environment and the dynamic control of ego vehicles using the cognitive safety area, the size of which is determined by naturalistic driving behavior. The artificial potential field method is used to comprehensively describe the influence of all external objects on the cognitive safety area, the lateral motion dynamics of which are determined by the attention mechanism of the human driver during lane changes. Then, the interaction between the designed cognitive safety area and the ego vehicle can be simplified into a spring-damping system, and the desired dynamic states of the ego vehicle can be obtained analytically for better computational efficiency. The effectiveness of this on improving traffic efficiency, driving comfort, safety, and real-time performance was validated using several comparative tests utilizing complicated scenarios with multiple vehicles.

## 1. Introduction

Since automated vehicles (AVs) can greatly improve traffic safety and efficiency and even reform the transportation modes of smart cities [1,2], they have been garnering increasing attention from academia, industry, business, and the government [3,4]. In addition to environmental perspectives, driving decisions and motion planning are also important parts of AVs [5,6], which generate control sequences according to detection results produced by surrounding objects. Since there are many methods of realizing object detection for camera [7,8], lidar [9,10], and fusion sensors [11,12], this study focuses on integrated driving decisions and motion planning.

Previously, since the driving scenarios faced by AVs were comparatively simple, rule-based methods, such as finite state machines, were widely used to decouple the driving task into multiple subtasks, e.g., car following [13], lane keeping, lane changing [14], emergency braking, and collision avoidance [15,16]. Now, for each driving subtask, a decision and control algorithm should be designed to enable safe and comfortable driving in specific driving scenarios. Such a framework has a clear and definite definition, is easily understood, and is beneficial for causality analysis of any problem with the products. Moreover, with increases in the intelligence of AVs, these vehicles are required to drive autonomously in increasingly complex traffic scenarios. It has become increasingly challenging to manually design rules for switching between different driving subtasks, since the engineer may struggle to identify all the possible traffic scenarios based on their engineering experience [17]. The designed system might malfunction, causing serious consequences when encountering unconsidered scenarios.

To overcome this challenge, an integrated structure was recently presented that accomplishes multiple driving tasks for a variety of scenarios using a unified algorithm [18]. According to the fundamental theory, there are two main strategies, i.e., optimization and learning. The optimization strategy converts local motion planning into an optimization problem by describing the driving requirements, vehicle dynamics, etc., as an objective function and equality and inequality constraints [3,17]. This strategy has the advantages of being easily understood and having a well-defined physical meaning and multi-objective optimization. However, numerical solving processes designed to find optimal solutions, such as nonlinear model predictive control (NMPC), dynamic programming, and quadratic programming, consume too many computation resources, especially when there are high nonlinearities or excessive non-convex constraints for safety [6]. The requirements for real-time performance are difficult to achieve using typical onboard processing units. Using linearization or approximation methods, the numerical process can be accelerated for some practical applications, but the optimization performance will be severely sacrificed, especially under dense or risky conditions with multiple surrounding objects [19].

Being different from the optimization strategy, the purpose of the learning strategy is to directly obtain the decision and planning algorithm from naturalistic or simulated driving data [20]. To capture and characterize the latent features of driving, neural networks such as long short-term memory networks and deep convolutional networks are widely used [21]. Differing from perception networks, the decision and planning network cannot be easily trained directly from a large amount of high-resolution and labeled data because of the closed-loop dynamics of motion planning systems. Even if the approximation error is very small, it will still be amplified by the feedback loop to an unacceptable level, sometimes so much so that the AV becomes unstable [22]. To ensure the closed loop has stability and the required driving performance, many methods based on reinforcement learning have been presented to train the policy network in a closed-loop simulation environment [4]. In addition to the gap between simulation and reality, the design of the reward function, to evaluate different driving tasks simultaneously, still presents a challenging task. Recently, researchers have tried to use professional driving data to guide the reinforcement learning process [22]. Moreover, 3D Gaussian splatting [23] and diffusion models [24] are some of the latest technologies to be revealed with great potential to realize high-fidelity training and promote the development of end-to-end driving networks and even real-world models [25] of AVs.

Unfortunately, explainability, interpretability, and reliability are critical barriers to the practical application of end-to-end networks [26], and using the modular structure is still a cost-effective and practical method for the current state of technology [27]. In this study, motivated by the fundamentals of artificial potential fields (APFs), an integrated decision and motion planning system is designed for highways with multiple objects moving in different directions. To simplify the design process, a two-layer structure is first presented to decouple the influence of the traffic environment and dynamic control of the ego vehicle (EV) within the cognitive safety area. Then, an APF is used to comprehensively describe the influence of all surrounding objects on the cognitive safety area, whose lateral motion dynamics are further determined by the attention mechanism of the human driver during a lane change. Finally, the interaction between the designed cognitive safety area and the EV can be simplified to a spring-damping system, and the desired dynamic states of the EV can be obtained analytically for better computational efficiency. The effectiveness is validated using both open- and closed-loop tests and comparison with other typical methods under several complicated scenarios from the HighD dataset. The results show that the proposed integrated driving decision and motion planning algorithm can achieve better comprehensive performance in terms of traffic efficiency, driving comfort, safety, and computational efficiency.

## 2. Two-Layered Decision and Planning Strategy

When there are multiple objects surrounding an AV, it is almost impossible for the engineer to identify all possible aspects of the static and dynamic relationships between them. However, this critical task forms the basis for designing rules to determine the current driving task and select the appropriate controller [18]. Considering the assumption of the APF, namely that driving behavior can be considered the interaction of various social forces in the traffic environment [28], provides a possible means of designing an integrated decision and motion planning system. Based on the potential of the APF and its application achievements [28,29,30], a two-layered structure, as shown in Figure 1, is designed to realize the integrated decision and planning with the following assumptions and considerations:(1)It is assumed that all vehicles obey the traffic rules of the highway;(2)There is a virtual area around the EV, which describes the required safety area for humans;(3)Considering the driving scenario on the highway, this virtual area is designed to be rectangular, and its width is set to be the width of the lane.

Though APFs have been successfully applied in a car-following scenario [29], pre-collision avoidance systems [30], etc., it is still a challenging task to design a unified field function that can describe the field forces generated from multiple objects in different directions for scenarios such as a lane change. In this study, by introducing the safety area, the complicated interaction between different objects is decomposed into two stages for simplification. One is between the surrounding vehicles and the safety area; the other is between the safety area and the EV. In the outer layer, i.e., the area marked by the gray background in Figure 1, only cognitive safety is considered, while the motion planning of vehicle dynamics is dealt with in the inner layer, which is represented by the light green background in Figure 1, and is referred to as the safety area in this study. The interaction between the EV and the surrounding vehicles in the outer layer is modeled with the APF, the design of the semi-static and dynamic parameters of which is discussed in Section 2.1 and Section 3, respectively. Since the potential field of the surrounding vehicles has already been considered in the outer layer, in the inner layer, only the relative motion between the EV and the safety area needs to be considered. Intuitively, the EV should stay in the center of the safety area when the system is static. The dynamics of the EV are determined by the degree of deviation from the center, which is designed according to previous studies and introduced in Section 2.2 in detail. Moreover, all variables and data are converted to the Frenet coordinates [31] in this study for the compatibility of the curve road of the highway.

### 2.1. Parameter Design of Safety Area

Aside from the width of the safety area, there are other parameters to design. Its length is determined using the required longitudinal space from the driver’s cognition of safety, which has been widely studied under car-following conditions using naturalistic driving data [13,14,28,29,30,32], and the desired distance in the intelligent driver model (IDM) proposed in [32] is used in this study to calculate the length of the safety area. A shorter following distance implies an aggressive driving style.

Comprehensively considering different driving requirements and running stability, the longitudinal speed of the safety region, $v_{x}$, is determined using
(1)$$v_{x}=\left\{\begin{array}{c} v_{p},\;\mathrm{Car}\;\mathrm{following}\;\mathrm{and}\;v_{p}\leq v_{d}\\ v_{d},\;\mathrm{Free}\;\mathrm{driving}\;\mathrm{or}\;v_{p}>v_{d} \end{array}\right.$$
where $v_{p}$ is the speed of the followed vehicle and $v_{d}$ is the desired speed set by the driver.

Under the lane-keeping condition, the safety region is designed to move along the lane, i.e., its lateral speed is zero. However, in the lane change scenario, its lateral dynamics become more complicated, and is studied in Section 3, for the following reasons:(1)The moment movement starts, and in which direction it starts, are directly related to the decision result for the lane change, which is affected by many factors, especially in the scenario where there are several surrounding vehicles are traveling in different directions [18,20];(2)The lateral dynamics of the safety area directly determine the lateral acceleration and, accordingly, the driving comfort of the EV;(3)Under the dynamical and critical conditions, the failure of the lane change should also be addressed using the designed method.

### 2.2. Motion Planning of EV

In this study, an EV is assumed to be governed by the spring-mass system in the safety area considering the following facts:(1)The influences from other surrounding objects have already been considered when designing the size and dynamics of the safety region in Section 2.1;(2)Previous studies of naturalistic driving have established that the vehicle acceleration varies with the distance and speed error almost linearly under stable driving conditions [13,30,32].

Based on the aforementioned considerations, an EV is controlled with [13,30,32]
(2)$$a_{x}=\frac{k_{1}\left(v_{x}-{\bar{v}}_{x}\right)+k_{2}(D-L)}{M},\;\;a_{y}=\frac{k_{3}\left({v_{y}-\bar{v}}_{y}\right)+k_{4}e_{y}}{M},$$
where $a_{x}$ and $a_{y}$ are the desired longitudinal and lateral accelerations of the EV, $v_{x}$ and $v_{y}$ are the longitudinal and lateral speeds of the safety area, ${\bar{v}}_{x}$ and ${\bar{v}}_{y}$ are the longitudinal and lateral speed of the EV, D is the distance between the preceding vehicle and the EV, L is the length of the safety area, $e_{y}$ is the lateral deviation of the EV from the center of the safety area, M is the mass of the EV, and $k_{i},\;i=1,\cdots ,4$ are the parameters that can be estimated using the naturalistic driving data under car-following and lane-keeping conditions [32,33].

## 3. Lateral Motion of Safety Area for Lane Change

In this section, the lateral dynamics of the safety area for a lane change are studied, including the design of the decision logic for the moving direction and the estimation of key parameters with naturalistic driving data.

### 3.1. Decision of Lateral Moving Based on Projection of Field Forces

The lateral movement decision of the safety area is similar to the lane-change decision of the human driver. In normal driving scenarios, the driver’s motivation for changing lanes is mostly to achieve a higher driving speed for better efficiency or drive in a safer lane [20,32,33]. For example, when a vehicle in front of an EV is driving at a low speed, the driver generally tends to change to the lane where the EV can drive at a higher speed. From this perspective, the driver’s intention to change lane can be considered as the result of the comprehensive effect caused by the social forces from the surrounding vehicles and the speed limit [28,29,30]. Based on the aforementioned analysis and considerations, the longitudinal social forces generated by the surrounding vehicles are projected toward the lateral direction of the EV to determine when and in which lateral direction the safety area should move, as shown in Figure 2. It should be noted that since the speed limit has already been considered in (1) when designing the safety area, the attraction force generated by it is ignored here.

As shown in Figure 2, the projected force along the lateral direction of the EV indicates the ability of the safety area to change to other lanes. Intuitively, when this projected force is larger than the social force of the lane, indicating the intention to stay in the current lane, the safety area will change lanes. Otherwise, it will stay in the current lane. The decision logic of the lateral motion of the safety area is designed to be:(3)$$\left\{\begin{array}{c} if\;F_{P}^{R}>F_{L}^{R}\;and\;F_{P}^{R}>F_{P}^{L},\;\;\;\;Change\;to\;the\;right\;lane\\ if\;F_{P}^{L}>F_{L}^{L}\;and\;F_{P}^{L}>F_{P}^{R},\;\;Change\;to\;the\;left\;lane\\ otherwise,\;\;\;\;Stay\;in\;the\;current\;lane \end{array}\right.$$
where $F_{L}^{\#}$ and $F_{P}^{\#}$ are the social force generated by the lane and the projected force for the lane change, respectively, and the superscript “#” denotes the lateral direction, i.e., left or right. These variables are calculated according to the APF as [13,30,32]:(4)$$F_{L}^{\#}=k_{3}v_{y}+k_{4}y,\;\;\;\;F_{P}^{\#}=\alpha \sum_{i\in \Omega^{\#}}F_{x,i}F_{x,i}=k_{1}\left(v_{x,i}-{\bar{v}}_{x}\right)+k_{2}e_{i}$$
where $v_{y}$ is the lateral velocity of the safety area, y is the lateral deviation of the safety area from the centerline, α is the conversion factor, $\Omega^{\#}$ is the set composed of the indexes of surrounding vehicles generating the projected force $F_{P}^{\#}$. The variables $F_{x,i}$, $v_{x,i}$, and $e_{i}$ are the social force, driving speed, and distance-tracking error of the nearby vehicle i, respectively. The parameters $k_{i},i=1,\cdots ,4$ are estimated from the naturalistic driving data under car-following and lane-keeping conditions [32,33].

### 3.2. Parameter Analysis Based on Naturalistic Driving Data

Though there are many studies on the parameter estimation of the APF based on naturalistic driving data, there are some new parameters in the moving decision logic proposed in Section 3.1, i.e., the conversion factor α in (4) and the moving speed from the current lane to the target. In this section, these parameters are analyzed using the HighD dataset [34], which is recorded by drone, and most data required for this study, such as vehicle states, relative dynamics, and trajectories, have already been pre-processed.

#### 3.2.1. Data Preparation

Since the unknown parameters of the safety area are related to lane change scenarios, the following rules are defined to select the required data from the HighD dataset:(1)The starting moment is the time when the EV leaves the centerline of the current lane;(2)The ending moment is the time when the EV reaches the centerline of the target lane;(3)The EV should move to the target lane continuously and smoothly without such conditions as driving in the lane for a long time;(4)The EV should move to the target lane successfully just in time, and the data of the lane change should also be complete;(5)The surrounding vehicles keep driving in their lanes without a lane change;(6)The data for scenarios where the EV moves to a lane with a lower speed and there is no vehicle in front of the EV are removed.

Based on the above rules, the typical lane change scenarios and statistical results are as shown in Figure 3, from which it can be found that the selected dataset covers a wide range of lane change scenarios.

#### 3.2.2. Statistical Analysis of Conversion Factor

According the decision logic of the lateral motion of the safety area described by (3) and (4), at the moment when the safety area starts to move, the following equation establishes the time at which the social forces vary continuously with the vehicle states:(5)$$F_{L}^{\#}=F_{P}^{\#}.$$

Since the selected scenarios in Section 3.2.1 are almost stable conditions before a lane change, the state of the safety area is assumed to be the same as that of the EV. Then, the conversion factor can be estimated using the data at the moment when the lane change starts, according to (4) and (5), as:(6)$$\alpha =\left(k_{3}{\bar{v}}_{y}+k_{4}e_{y}\right)/\sum_{i\in \Omega^{\#}}F_{x,i}.$$

Using (6), the statistical result of α under different lane-change scenarios is shown in Figure 4. The following can be seen:(1)A bigger conversion factor means that less longitudinal social force can be projected toward the lateral direction. This implies that the EV tends to stay in the current lane, i.e., there is less intention to change lanes;(2)With the increase in EV speed, its intention to change lanes increases slightly and the average value of the conversion factor decreases from about −0.21 to −0.35;(3)For a given speed, the converting factor varies in a comparatively large range, since there are obvious differences in driving style for different drivers [35]. Accordingly, the conversion factor can act as a customization parameter to be set by the user, and a lower one implies an aggressive driving style.

#### 3.2.3. Statistical Analysis of Moving Speed

Since an EV is controlled by the spring-mass system described by (2), which deals with the dynamical behavior of an EV during a lane change, the safety area is assumed to move uniformly in a lateral motion in the scenario of the lane change. The statistical results of the average time required for a lane change under different speeds of an EV are shown in Figure 5.

From Figure 5, the following can be found:(1)A shorter time for a lane change means a more aggressive driving style;(2)The average time required for a lane change varies with the speed of the EV in a small range from about 3.5 s to 4.5 s; therefore, in this study, the influence of speed on the time for a lane change is ignored;(3)For a given speed of an EV, the time for a lane change varies in a large range. Being similar to the conversion factor, this may be caused by the style of different drivers;(4)Accordingly, the required time for a lane change can also act as a customization parameter to be set by the user.

## 4. Validation and Analysis

To demonstrate the effectiveness and performance of the proposed strategy, it was validated with open-loop and closed-loop tests under different traffic conditions by comparing it with other widely used methods, i.e., NMPC [3] and IDM [32]. The former is a representative method of optimization strategy. The parameters of the latter were optimized with the same dataset used in this study, i.e., HighD. Essentially, it is also a learning-based method. The pseudocode of the proposed algorithm is denoted as IDP in the following (Algorithm 1).
**Algorithm 1. IDP: Integrated driving decision and motion planning algorithm****Initialization**: desired driving speed vd, converting factor α, lateral moving speed, parameters of following distance in [33], parameters of motion dynamics in (2);**While** EV drives autonomously  Obtain detection results of surrounding vehicles and lanes;  Set longitudinal speed of safety area using (1);  Calculate social force generated by lane and projected force for lane change using (4);  Determine direction of lane change using (3);  **If** EV stays in current lane    Set lateral deviation of safety area to be zero;  **Otherwise, if** EV makes lane change    Calculate lateral deviation of safety area according to lateral moving speed;  End  Calculate desired longitudinal and lateral acceleration of EV using (2);  Send desired control values to actuator controllers;**End**


### 4.1. Open-Loop Tests Based on Naturalistic Driving Data

Since both the IDM [32] and the proposed method learn from naturalistic driving data, they are first tested using the dataset for the lane change scenario obtained in Section 3.2.1 to show the advantage of the proposed method in suiting naturalistic driving behavior. The relative distance and speed between the EV and surrounding objects and other required signals are taken as the input. The state of the EV is controlled using the tested methods according to information about the environment. The lateral acceleration, lateral velocity, and lateral displacement of the EV are compared with the original data to evaluate the performance, and the comparative results of the state error of the EV and the time error of the moment of the lane change are shown in Table 1 and Table 2, respectively.

Overall, the fitting accuracy of the proposed method is much better than that of the IDM, which implies that the EV controlled with the proposed method more closely resembles a human driver. The delay time of the lane change moment can reflect the estimation accuracy of the intention to change lane. As shown in Table 2, compared with the IDM, the proposed method has better accuracy. Moreover, as discussed in Section 3.2.2, the strength of the intention to change lane can be adjusted with the conversion factor α to make the system to adapt to different requirements of driving style.

### 4.2. Closed-Loop Validation in Naturalistic Driving Scenarios

Since the decision and motion planning system of AVs works in a closed loop composed of AVs and traffic environments, the open-loop tests described in Section 4.1 cannot evaluate performance comprehensively, accurately, and objectively. In this section, the lane-changing scenarios selected from the HighD dataset in Section 3.2.1 are further used to test the proposed method using a simulation. The simulation system is composed of Prescan and Matlab [36,37]. The tested algorithm and the vehicle dynamics model of the EV are run in Matlab/Simulink and the information is exchanged via the synchronous transmission interface provided by Prescan, which also simulates the traffic scenario.

To evaluate the proposed method comprehensively, several indexes, including safety, efficiency, and comfort, are adopted. Comfort performance is measured with the weighted root mean square value of acceleration defined in the international standard ISO-2631-1 [38]. Safety and efficiency performances are measured using the following [39]:(7)$$J_{s}=\underset{i}{\min}\frac{min(D_{i},120)}{{\bar{v}}_{x}},\;\;\;\;J_{e}=10tanh(\frac{1.83{\bar{v}}_{x}}{\min\left(v_{p},v_{l}\right)}).$$
where $J_{s}$ and $J_{e}$ are the indexes for safety and efficiency, respectively; $D_{i}$ is the relative distance between the EV and vehicle i; ${\bar{v}}_{x}$ and $v_{p}$ are the speed of the EV and the followed vehicle, respectively; and $v_{l}$ is the speed limit.

More than 20 lane-change scenarios were selected randomly from the dataset obtained in Section 3.2.1 to conduct the simulation tests. The proposed method was validated by comparing with both NMPC [3] and IDM [32], and a violin plot of the evaluation indexes is shown in Figure 6.

In Figure 6, the proposed method is denoted as IDP. The tested algorithms can all control the EV without a collision, but comparatively, the EV controlled using NMPC behaves more conservatively (see Figure 6a) and uncomfortably (see Figure 6c). This is because the parameters of IDP and IDM were obtained using naturalistic driving data, so they control the EV more similarly to a human driver. Comparatively, the control value of the NMPC is obtained using numerical optimization, which can only consider the driving style indirectly by adding the corresponding indexes in the objective function or constraint equations. Moreover, the computation resources required by the NMPC are much larger, since the interactive process for numerical optimization consumes much more computation resources, which also increases exponentially with the dimensions of optimization variables [3]. Its average period is about 80 ms with the simulation platform (Intel(R) Core (TM) i7-10750H CPU @ 2.60 GHz, RAM 16 GB, and NVIDIA GeForce GTX 1650 Ti GPU, Beijing Jingwei Hirain Technology Co., Ltd., Beijing, China), which is more than seven times that of IDP and IDM. Furthermore, by comparing IDP and IDM, it can be found that (1) the overall comfort index of IDP is somewhat larger than that of the IDM (see Figure 6c); (2) the average efficiency index of IDP is therefore somewhat better than that of the IDM (see Figure 6b); and (3) the overall safety index of IDP is better than that of the IDM (Figure 6a). In summary, IDP has a better overall performance than IDP and the NMPC.

### 4.3. Test Under Multi-Task and Dynamic Scenario

To further validate the proposed method, the following test scenario, shown in Figure 7, is designed. In this scenario, the surrounding vehicles run dynamically, and the EV is required to accomplish multiple driving tasks. As shown in Figure 7, V1–V5 move in a straight line with constant speeds, and V6 makes a lane change from the left to the middle while decelerating to 90 km/h with a deceleration of −2 m/s^2^. To pass in this scenario, the EV must make a lane change to avoid a collision with V6; it then needs to undertake a series of subtasks, such as lane keeping and collision avoidance, to achieve the final driving speed. The initial speed and relative position are depicted in Figure 7.

The comparative test results are shown in Figure 8. All three methods can pass the test without collision (see Figure 8a). During the first 100 m, the EV makes a lane change to avoid a collision with V6, which changes to its right lane and decelerates simultaneously. By comparing the risk of three lanes, the EV decides to change to the lane to its left. Then, the EV follows V5 until the EV overtakes V4. After the second lane change between about 400 and 500 m, the EV controlled by IDP is somewhat faster than the others. Moreover, it can be found from the dynamic states of the EV, shown in Figure 8b–e, that the EV controlled using IDP drives in a more stable manner, implying that it can provide more comfortable driving experience.

## 5. Conclusions

To realize integrated decision and motion planning for highways, this study presents a two-layered structure to decouple the complex interaction between different objects, and the key parameters of the artificial field function are designed and analyzed using naturalistic driving data. The analysis and validation results show the following:(1)The proposed framework can decompose the complex interactions in driving scenarios and motion planning tasks for an EV by using the safety area;(2)The proposed method has good adaptability to different scenarios by integrating the motion planning algorithm under different conditions into a unified spring-mass system, which also benefits performance in terms of comfort and smoothness;(3)The open/closed-loop test results show that the proposed method demonstrates better overall performance than IDP and NMPC and can control the EV safely, efficiently, and comfortably in real time.

The following open questions need to be studied further and validated:(1)Ethical and legal requirements must be satisfied for autonomous driving, especially for unavoidable accident scenarios [40]. How to simultaneously consider such requirements in the design of the driving decision and motion planning system needs to be further studied.(2)As shown in Figure 3a, though the HighD dataset includes various highway scenarios, the proposed method needs to be tested using more datasets and considering additional factors, such as weather, to show the robustness and generalization of the method.

## Figures and Tables

**Figure 1 sensors-25-00026-f001:**
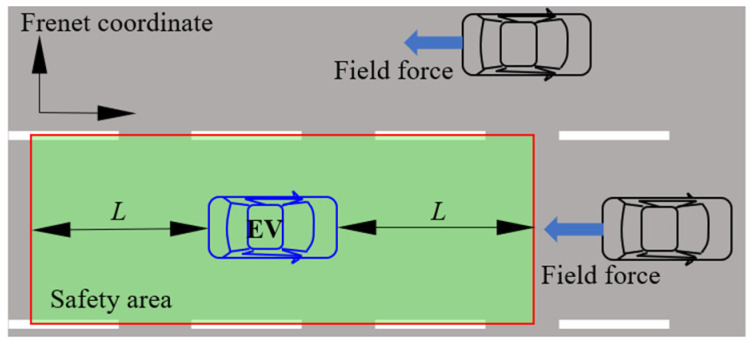
Fundamental aspects of two-layered decision and planning system.

**Figure 2 sensors-25-00026-f002:**
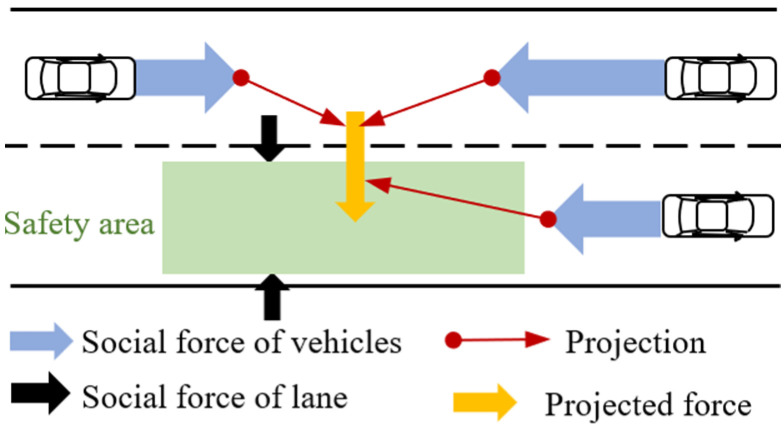
Decision strategy of lateral motion.

**Figure 3 sensors-25-00026-f003:**
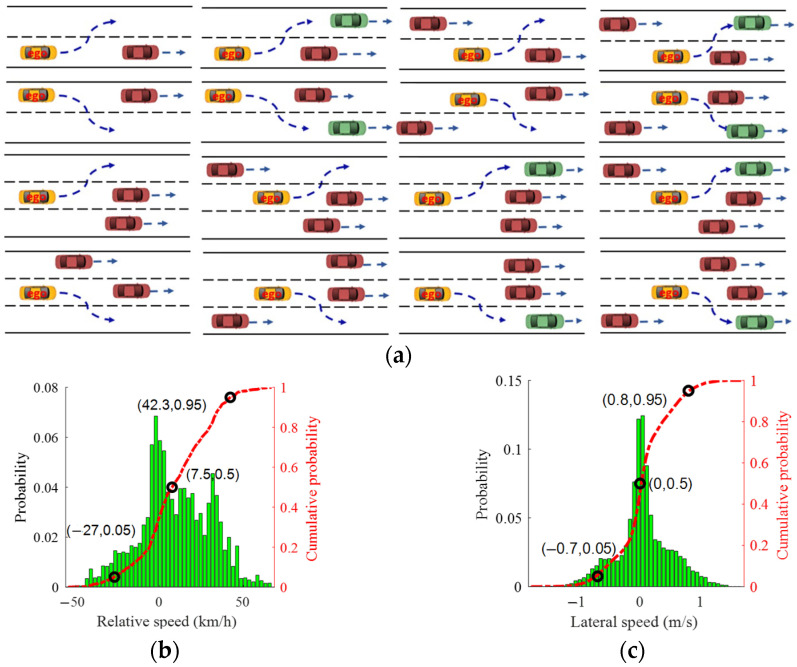
Results of selected lane change dataset: (**a**) typical lane change scenarios; (**b**) distribution of relative speed; (**c**) distribution of lateral speed of EV.

**Figure 4 sensors-25-00026-f004:**
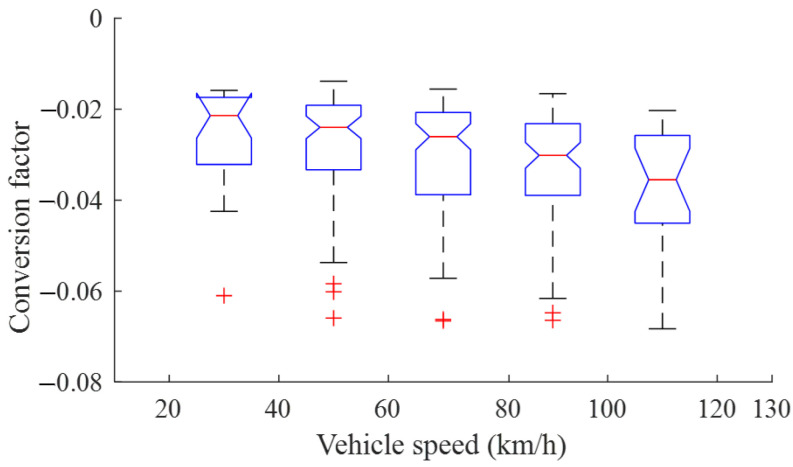
Statistical results of converting factor α.

**Figure 5 sensors-25-00026-f005:**
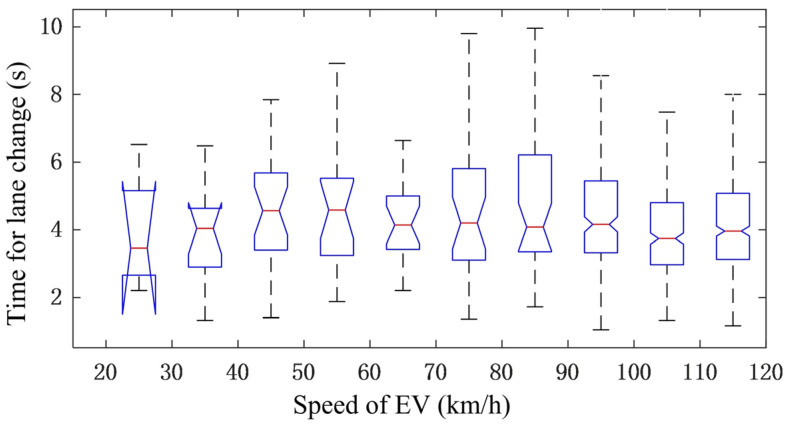
Statistical results of lane change time.

**Figure 6 sensors-25-00026-f006:**
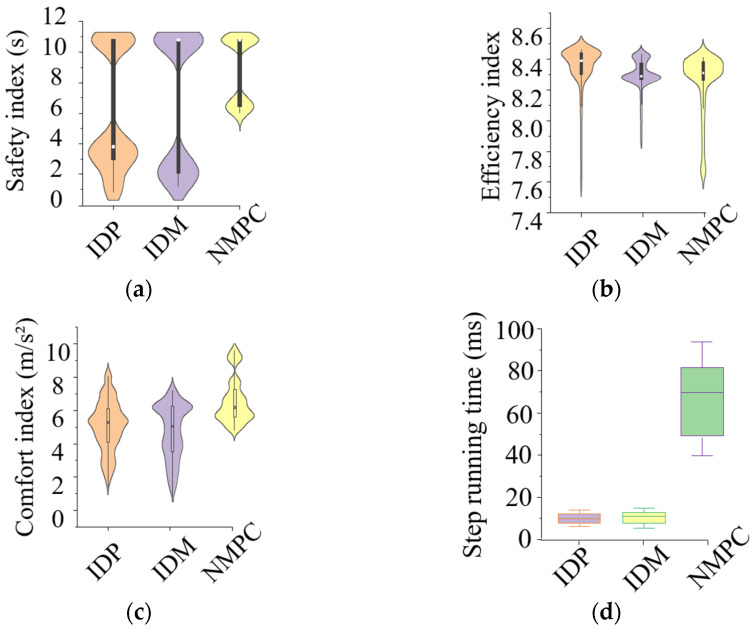
Statistical results of evaluation indexes: (**a**) safety performance; (**b**) efficiency performance; (**c**) comfort performance; (**d**) running time.

**Figure 7 sensors-25-00026-f007:**
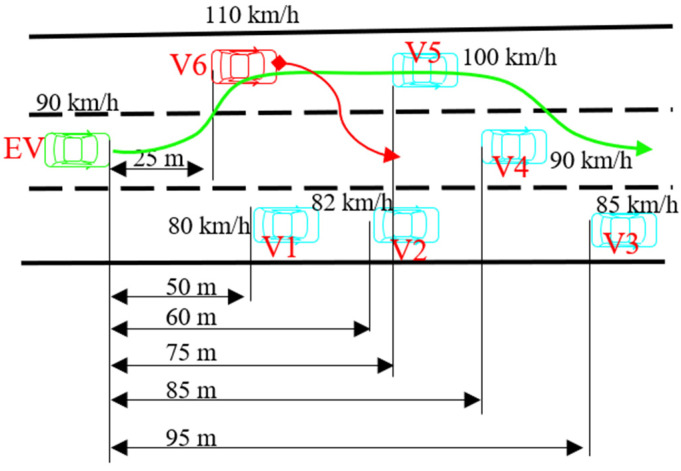
Diagram of dynamic multi-task scenario.

**Figure 8 sensors-25-00026-f008:**
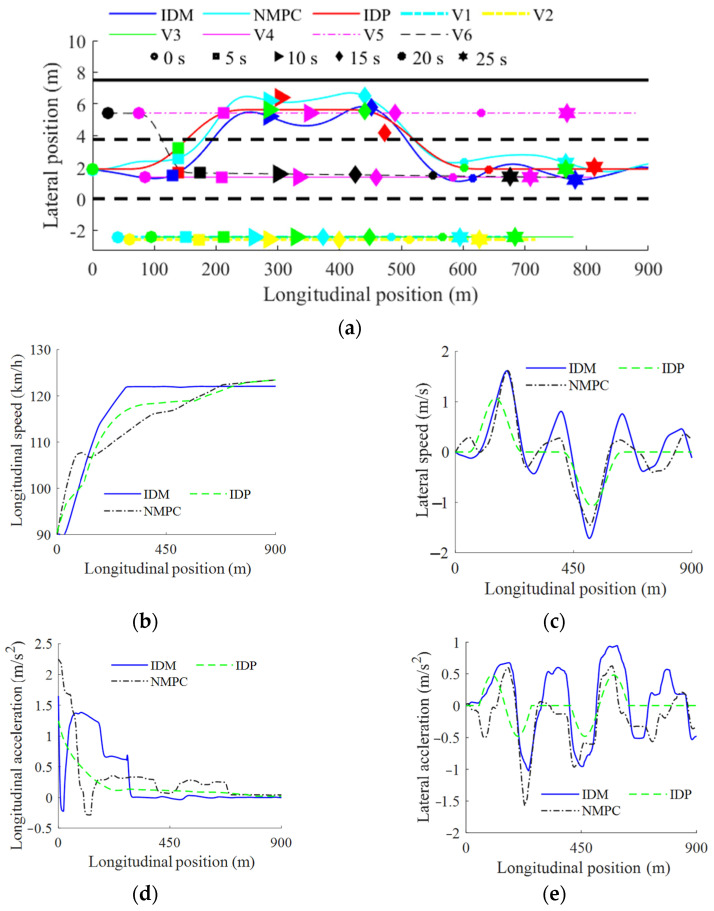
Results in multi-task and dynamic scenario: (**a**) bird’s-eye view of trajectory; (**b**) longitudinal speed; (**c**) lateral speed; (**d**) longitudinal acceleration; (**e**) lateral acceleration.

**Table 1 sensors-25-00026-t001:** Comparative results of state error.

	Proposed Method			IDM		
	Acceleration (m/s^2^)	Speed (m/s)	Displacement (m)	Acceleration (m/s^2^)	Speed (m/s)	Displacement (m)
Mean	−0.011	−0.059	−0.061	−0.168	−0.225	0.547
Variance	0.003	0.008	0.064	0.052	0.400	0.769

**Table 2 sensors-25-00026-t002:** Comparative results of time error.

	Mean (s)	Variance (s)	Max (s)	Min (s)
Proposed method	−0.450	1.632	1.660	−3.641
IDM	−1.969	2.259	0.950	−5.389

## Data Availability

The data can be made available by the corresponding author on a contact basis.

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
