# Peer review of "Integrated Decision and Motion Planning for Highways with Multiple Objects Using a Naturalistic Driving Study"

_sensors, 2024, doi:10.3390/s25010026_

Round 1

Reviewer 1 Report

Comments and Suggestions for Authors

(1) The manuscript presents valuable research on autonomous vehicle decision-making and motion planning, but requires several improvements to meet publication standards.

(2) The two-layer structure model needs more detailed explanation of its inner workings to enhance interpretability and transparency.

(3) The reference profile is robust, comprising 30 citations with 93.4% from the past decade (66.7% within five years), demonstrating strong currency in the literature, but insufficient.

(4) The authors may incorporate more CV articles for the integrity of the manuscript (Dual-Frequency Lidar for Compressed Sensing 3D Imaging Based on All-Phase Fast Fourier Transform; Journal of Optics and Photonics Research. 3D vision technologies for a self-developed structural external crack damage recognition robot; Automation in Construction. Path planning for mobile robots in unstructured orchard environments: An improved kinematically constrained bi-directional RRT approach; Computers and Electronics in Agriculture.).

(5) The method's generalization capabilities need to be better demonstrated across different regions, traffic conditions, and complex urban environments. Additionally, real-time performance metrics on actual vehicles with limited computing resources should be provided.

(6) The paper should address ethical and legal considerations in autonomous driving decision-making, particularly regarding unavoidable accident scenarios.

(7) The study would benefit from using multiple datasets beyond HighD to enhance model robustness and generalization. Consider incorporating additional environmental factors such as weather and road conditions.

(8) The experimental results section requires more rigorous statistical analysis, including confidence intervals and ANOVA. The comparative analysis should extend beyond performance metrics to include algorithm complexity and computational resource requirements.

(9) The limitations of the study should be clearly discussed, including methodological constraints and dataset limitations. Future research directions and practical applications should be more thoroughly explored.

(10) The paper should include explanations of key technical terms and concepts in the introduction to improve readability.

(11) System performance in extreme or rare traffic situations (emergency scenarios, extreme weather) needs more detailed discussion.

(12) The process of adapting to different driving styles through parameter tuning should be better explained, particularly regarding safety and effectiveness maintenance.

Author Response

Please find the response in the attachment.

Reviewer 2 Report

Comments and Suggestions for Authors

This paper is inspired by the principles of artificial potential fields (APFs), which is an integrated system for making decisions and planning movements on highways with various objects in different directions. It should be noted that APFs are mathematical functions used in robotics to control the movement of autonomous vehicles (AVs), for example. The APFs simulate the effects of attractive and repulsive forces on the decision-making systems of human vehicle drivers or AVs' control systems.  In the paper, there is proposed a two-layered structure that separates the influence of the traffic environment and the dynamic control of the ego vehicle (EV) by the cognitive safety area. The APF (Adaptive Perceptual Field) is used to describe the impact of all surrounding objects on the cognitive safety zone. The lateral dynamics of this safety zone are determined by the attention mechanisms of human drivers during lane changes. Then, the interaction between the safety zone and the ego vehicle was modeled as a spring-damper system. This allowed us to analytically determine the desired dynamic states of the vehicle, leading to improved computational efficiency. The effectiveness of the system was validated through simulations. 

The paper is well-organized and contains some interesting new insights. The chosen topic is highly relevant, and the methods used are impressive. Artificial potential fields are a powerful tool for simulating the behavior of road users, partially based on models of pedestrian dynamics that take into account social forces. However, this paper could be further improved. The following specific areas require attention:

1. The introduction should be improved. The article would have become more valuable if the authors had more clearly formulated the goals and main advantages of their approach. These benefits are associated with improved traffic, increased vehicle maneuverability, and reduced travel time for motorists and other road users. What other potential benefits might there be to the methods proposed by the authors, other than improving traffic safety?

2. The literature review could be improved. In this article, we focus on improving traffic safety. At the same time, there are several significant works in the field that focus on agent-based modeling and simulation-based optimization techniques for traffic improvement. To this end, it would be helpful to mention and cite some of these articles, especially the following:

[1] Akopov A.S., Beklaryan L.A, and Beklaryan A.L. Simulation-Based Optimisation for Autonomous Transportation Systems Using a Parallel Real-Coded Genetic Algorithm with Scalable Nonuniform Mutation. Cybernetics and Information Technologies. 21, 3 (Sep 2021), 127–144. https://doi.org/10.2478/cait-2021-0034.

[2]  Alqurashi, R.; Altman, T. Hierarchical Agent-Based Modeling for Improved Traffic Routing. Appl. Sci. 2019, 9, 4376. https://doi.org/10.3390/app9204376

3. There is no clear, algorithmic description of the motion planning algorithm proposed in the article, which is often mentioned in the paper. In particular, the Conclusion contains the following sentence "The proposed method has fine adaptability to different scenarios by integrating the motion planning algorithm under different conditions into a unified spring-mass system team, which is also beneficial to the comfort and smoothness performance; "

This statement should be illustrated using either a flowchart or pseudocode, in relation to the motion planning algorithm proposed by the authors.

Author Response

(The authors gave the same response as above.)

Round 2

Reviewer 2 Report

Comments and Suggestions for Authors

The paper has undergone significant revisions based on the reviewers' comments. The research provides a valuable contribution to the field of traffic management, specifically addressing issues related to traffic efficiency, driver comfort and safety, and real-time performance. As a result of these revisions, the paper has been significantly improved and is now highly recommended for publication.